# Metabolomic Analysis of Trehalose Alleviating Oxidative Stress in Myoblasts

**DOI:** 10.3390/ijms241713346

**Published:** 2023-08-28

**Authors:** Shuya Zhang, Xu Qiu, Yue Zhang, Caihua Huang, Donghai Lin

**Affiliations:** 1Key Laboratory of Chemical Biology of Fujian Province, MOE Key Laboratory of Spectrochemical Analysis and Instrumentation, College of Chemistry and Chemical Engineering, Xiamen University, Xiamen 361005, China; syzhang@stu.xmu.edu.cn (S.Z.); qiuxu@stu.xmu.edu.cn (X.Q.); 20520211152128@stu.xmu.edu.cn (Y.Z.); 2Research and Communication Center of Exercise and Health, Xiamen University of Technology, Xiamen 361021, China; huangcaihua@xmut.edu.cn

**Keywords:** trehalose, oxidative stress, skeletal muscle, Keap1-Nrf2, NMR-based metabolomics

## Abstract

Trehalose, a naturally occurring non-toxic disaccharide, has attracted considerable attention for its potential in alleviating oxidative stress in skeletal muscle. In this study, our aim was to elucidate the metabolic mechanisms underlying the protective effects of trehalose against hydrogen peroxide (H_2_O_2_)-induced oxidative stress in C2C12 myoblasts. Our results show that both trehalose treatment and pretreatment effectively alleviate the H_2_O_2_-induced decrease in cell viability, reduce intracellular reactive oxygen species (ROS), and attenuate lipid peroxidation. Furthermore, using NMR-based metabolomics analysis, we observed that trehalose treatment and pretreatment modulate the metabolic profile of myoblasts, specifically regulating oxidant metabolism and amino acid metabolism, contributing to their protective effects against oxidative stress. Importantly, our results reveal that trehalose treatment and pretreatment upregulate the expression levels of P62 and Nrf2 proteins, thereby activating the Nrf2-NQO1 axis and effectively reducing oxidative stress. These significant findings highlight the potential of trehalose supplementation as a promising and effective strategy for alleviating oxidative stress in skeletal muscle and provide valuable insights into its potential therapeutic applications.

## 1. Introduction

Oxidative stress, stemming from the presence of reactive oxygen intermediates [1], disrupts redox signaling and control, thereby leading to various damage and diseases within the body [2,3]. Reactive oxygen species (ROS), encompassing the superoxide anion (O•^−^), hydroxyl radical (HO•), and hydrogen peroxide (H_2_O_2_) [4], cause harm to proteins, nucleic acids, and cell membranes [1]. The complex relationship between ROS production and skeletal muscle function and well-being is widely recognized [5,6]. While reactive oxygen species (ROS) play a crucial role in regulating intracellular signal transduction at appropriate levels, excessive ROS generated from over-exercising can lead to severe muscle damage [7,8,9]. Despite extensive research into skeletal muscle oxidative stress in recent years, a definitive solution to alleviate this stress remains elusive [10].

The Keap1-Nrf2 system is widely recognized as the primary defense mechanism against oxidative stress [11]. Keap1, a protein abundant in thiols, possesses a unique arrangement of cysteine residues that enables it to sense H_2_O_2_ [12]. Normally, Keap1 binds to Nrf2 in the cytoplasm and regulates its activity [13]. However, when exposed to ROS or electrophiles, Keap1 undergoes modifications that disrupt its interaction with Nrf2, leading to the translocation of Nrf2 into the nucleus [11,14]. Once translocated into the nucleus, Nrf2 forms a complex with sMAF and binds to the antioxidant response element (ARE), thereby initiating the transcription of a range of antioxidative and cytoprotective proteins, such as NQO1, HO-1, CAT, SOD, and GSH-Px [15,16,17].

Trehalose, a disaccharide consisting of two glucose molecules linked by a 1,1-glycosidic bond, is found in many organisms, including bacteria, plants, and selected animals, where it performs diverse functions such as protecting proteins and membranes from desiccation and acting as a structural component of bacterial cell walls [18]. In addition, trehalose has been shown to affect biological processes in mammals with minimal side effects [19], including its involvement in autophagy [20,21] and its anti-cancer properties [22,23]. Studies have also revealed the antioxidant properties of trehalose, including the reduction in H_2_O_2_-induced ROS accumulation, the preservation of CAT and SOD activity [24,25], the scavenging of ROS, and the reduction in lipid peroxide levels in bovine milk [26]. However, the role of trehalose in myoblast oxidative stress and the underlying mechanisms remain to be elucidated.

In recent years, metabolomics has experienced considerable progresses, going from the identification of biomarkers to revealing the underlying mechanisms of phenotypes [27,28,29]. Among the various analytical approaches available, NMR-based metabolomics has emerged as a prominent method for metabolic profiling, owing to its numerous advantages [30]. In this study, we performed NMR-based metabolomic analysis to explore the potential of trehalose in alleviating oxidative stress in skeletal muscle. In particular, we investigated the therapeutic effect of trehalose treatment and the preventive effect of trehalose pretreatment on H_2_O_2_-exposed C2C12 myoblasts. Our results showed that both trehalose treatment and pretreatment effectively attenuated the H_2_O_2_-induced decrease in cell viability, reduced intracellular ROS and lipid peroxidation, and partially mitigated metabolic alterations associated with oxidative stress. This study provides novel insights into the antioxidant capacity of trehalose in alleviating oxidative stress in skeletal muscle, contributing to our understanding of its potential therapeutic applications.

## 2. Results

In this study, we used two interventional approaches to investigate the protective effects of trehalose on C2C12 myoblasts under oxidative stress (Appendix A). We comprehensively analyzed cell phenotypes, cellular metabolism, and protein expression to elucidate the metabolic mechanisms underlying the therapeutic effect of trehalose treatment on cells with short-term exposure to H_2_O_2_, as well as the preventive effect of trehalose pretreatment on cells with prolonged exposure to H_2_O_2_.

For the trehalose treatment experiment (Appendix A), we divided the C2C12 cells into four groups (H, HT, C, T): The H cells were cultured in DMEM supplemented with H_2_O_2_ for 2 h and then continuously cultured in DMEM for 24 h, while the HT cells were cultured in DMEM supplemented with H_2_O_2_ for 2 h and then continuously cultured in DMEM supplemented with trehalose for 24 h. In addition, the C and T cells were cultured in DMEM or DMEM supplemented with trehalose, respectively, without exposure to H_2_O_2_ for the entire duration of 24 h.

For the trehalose pretreatment experiment (Appendix A), we utilized three groups (pTH, pH, pC) of C2C12 cells: The pTH cells were cultured in DEME supplemented with trehalose for 12 h and then continuously cultured in DEME supplemented with H_2_O_2_ and trehalose for 12 h, followed by an additional 24 h of culture. The pH cells were cultured in DEME for 12 h and then continuously cultured in DEME supplemented with H_2_O_2_ for 12 h, followed by an additional 24 h of culture. Lastly, the pC cells were cultured in DEME for 12 h and then continuously cultured in DEME for 12 h, followed by an additional 24 h of culture, without exposure to H_2_O_2_.

### 2.1. Trehalose Attenuated H_2_O_2_-Induced Decrease in Myoblast Viability

We used the MTS assay to establish the optimal concentration of trehalose for our oxidative stress model using C2C12 cells exposed to 200 µM H_2_O_2_ [31,32]. Our results indicated that trehalose concentrations up to 10 mM did not exhibit any toxicity to the myoblasts (Figure 1A). Moreover, myoblast viability was significantly improved when trehalose concentrations reached 10 mM or higher in the presence of H_2_O_2_ (Figure 1C). Consequently, we selected 10 mM trehalose for subsequent experiments.

In our evaluation of the therapeutic and preventive effects of trehalose on cells exposed to oxidative stress, we primarily focused on cell phenotypes (Appendix A). Following exposure to H_2_O_2_, both cell viabilities (Figure 1C,D) and cell densities (Figure 1B,E) experienced substantial reductions when compared to the control (H vs. C and pH vs. pC). Trehalose supplementation, however, reversed the changing trends (HT vs. H and pTH vs. pH), demonstrating its ability to attenuate the H_2_O_2_-induced decrease in cell viability and thus protect myoblasts. 

### 2.2. Trehalose Facilitated ROS Scavenging in Myoblasts

The effect of ROS on cell fate is well established [4], and their levels can be monitored using fluorescence probes [33,34]. Our research showed that the fluorescent intensities in the H and pH cells were higher compared to the C and pC cells (Figure 2A,B,D,E), indicating an abnormal accumulation of ROS in myoblasts after exposure to H_2_O_2_. Interestingly, the fluorescence intensities in the HT and pTH cells were lower than those in the H and pH cells (Figure 2A,B,D,E), indicating that trehalose aided in the scavenging of ROS in cells. This observation was further supported by the significant increase in MDA levels following exposure to H_2_O_2_ (H vs. C and pH vs. pC) and the subsequent decrease upon trehalose treatment (HT vs. H, Figure 2C) and pretreatment (pTH vs. pH, Figure 2F). These results highlight the ability of trehalose supplementation to reduce ROS levels and H_2_O_2_-induced lipid peroxidation in cells under oxidative stress.

### 2.3. Trehalose Altered the Metabolic Profile of Myoblasts

To investigate the effects of trehalose on myoblasts under oxidative stress from the perspective of metabolic regulation, we recorded 1D ^1^H-NMR spectra of the C, T, H, and HT cells (Figure 3A) as well as the pC, pH, and pTH cells (Figure 3B). Using the Chenomx NMR Suite software (Version 8.3), the Human Metabolomics Database (HMDB), and relevant literature, we identified 37 (Appendix A) and 34 (Appendix A) metabolites based on the NMR spectra (Figure 3A,B), respectively. As expected, trehalose was only detected in the T, HT, and pTH cells (Appendix A) due to its supplementation in the culture medium. The resonance assignments of the metabolites were verified using 2D ^1^H-^13^C HSQC (Appendix A) and 2D ^1^H-^1^H TOCSY spectra (Appendix A).

Unsupervised principal component analysis (PCA) was used to examine the metabolic profiles of myoblasts in different groups, with trehalose-related NMR data excluded from the multivariate data analysis to avoid potential confounding. The PCA score plot of the C, T, H, and HT groups of cells illustrates the grouping trends of the metabolic profiles of these four groups for the trehalose treatment experiment (Figure 4A). Furthermore, the PCA score plots for H vs. C, HT vs. H, and T vs. C revealed a clear metabolic distinction among these four groups (Figure 4B,D,F). Similarly, the PCA score plot of the pC, pH, and pTH groups of cells also displays the grouping trends of the metabolic profiles of these three groups for the trehalose pretreatment experiment (Figure 4H), and PCA scores plots for pH vs. pC and pTH vs. pH revealed a clear metabolic distinction among these three groups (Figure 4I,K).

These results indicated that both short-term and long-term exposure to H_2_O_2_ resulted in changes in the metabolic profile of cells. Furthermore, trehalose supplementation before and after oxidative stress, as well as trehalose supplementation under normal culture conditions, significantly altered the metabolic profile of cells.

To maximize the differentiation of metabolic profiles between groups of cells, we conducted supervised orthogonal partial least squares-discriminant analysis (OPLS-DA) based on the NMR dataset. The OPLS-DA scores plots shown in Figure 4 clearly illustrated the metabolic differentiation between the H and C groups (Figure 4C), the HT and H groups (Figure 4E), the T and C groups (Figure 4G), the pH and pC groups (Figure 4J), and the pTH and pH groups (Figure 4L). In addition, response permutation tests with 200 cycles confirmed the reliability of the OPLS-DA models (Appendix A). 

### 2.4. Identification of Significant Metabolites in Myoblasts

We utilized the criterion of variable importance in projection (VIP) >1 obtained from the OPLS-DA models (Figure 5A–E, Appendix A) to identify significant metabolites that primarily contributed to the metabolic separation between groups of myoblasts. For the therapeutic effect of trehalose (Appendix A), we identified 15, 13, and 15 significant metabolites from pairwise comparisons of H vs. C (Figure 5A), HT vs. H (Figure 5B), and T vs. C (Figure 5C), respectively. For the preventive effect of trehalose (Appendix A), we identified significant 17 and 12 metabolites from pairwise comparisons of pH vs. pC (Figure 5D) and pTH vs. pH (Figure 5E), respectively. Notably, six significant metabolites were shared between the pairwise comparisons of H vs. C and HT vs. H, including taurine, glutamine, formate, choline, dimethylamine, and fucose (Appendix A). Similarly, seven significant metabolites were shared between the pairwise comparisons of pH vs. pC and pTH vs. pH, including glutamate, valine, creatine, tyrosine, leucine, isoleucine, and PC (O-phosphocholine) (Appendix A).

### 2.5. Trehalose Changed Metabolites Levels in Myoblasts

To further compare the relative levels of the identified metabolites (except trehalose) between groups of cells, we conducted univariate statistical analyses of their relative NMR integrals (Table 1 and Table 2). Using the criterion of *p* < 0.05, We identified 12, 7, and 9 differential metabolites from pairwise comparisons of H vs. C, HT vs. H, and T vs. C, respectively (Appendix A). Notably, three differential metabolites (taurine, glutamine, and formate) exhibited opposite changing trends when comparing H vs. C and HT vs. H, indicating that trehalose partially reversed the impairment caused by oxidative stress. Similarly, we identified 12 and 10 differential metabolites from pairwise comparisons of pH vs. pC and pTH vs. pH, respectively (Appendix A). Among these, three differential metabolites (valine, PC, and glutamate) were shared between the two sets of comparisons. Valine and PC exhibited opposite changing trends, while the levels of glutamate did not change.

### 2.6. Trehalose Altered Metabolic Pathways in Myoblasts

To gain further insights into the metabolic regulation of trehalose in myoblasts, we identified significantly altered metabolic pathways, referred to as “significant pathways,” using the criteria of pathway impact value (PIV) > 2 and *p* < 0.05 (Appendix A). In the pairwise comparisons of H vs. C and HT vs. H, we identified a total of five and six significant pathways, respectively. Among these, four significant pathways were shared by the two sets of comparisons: P1: Alanine, aspartate, and glutamate metabolism; P3: Taurine and hypotaurine metabolism; P4: beta-Alanine metabolism; P6: Glycine, serine, and threonine metabolism (Figure 6A,B and Appendix A). Additionally, we identified five metabolic pathways significantly altered by trehalose supplementation in normal medium (T vs. C): P1: Alanine, aspartate, and glutamate metabolism; P2: D-Glutamine and D-glutamate metabolism; P3: Taurine and hypotaurine metabolism; P5: Glutathione metabolism; P7: Histidine metabolism (Figure 6C and Appendix A). 

Furthermore, the pairwise comparisons of pH vs. pC and pTH vs. pH revealed seven significant pathways. Among these, seven significant pathways were shared by the two sets of comparisons: P1: Alanine, aspartate, and glutamate metabolism; P2: D-Glutamine and D-glutamate metabolism; P5: Glutathione metabolism; P6: Glycine, serine, and threonine metabolism; P8: Phenylalanine, tyrosine, and tryptophan biosynthesis; P9: Phenylalanine metabolism (Figure 6D,E and Appendix A).

To illustrate the metabolic regulation of trehalose supplementation and H_2_O_2_ exposure in myoblasts, we generated overviews that encompassed relevant pathways and essential metabolites from the Kyoto Encyclopedia of Genes and Genomes (KEGG) database and MetaboAnalyst 5.0 webserver (Appendix A). These finds indicated that trehalose can protect myoblast from oxidative stress by altering the metabolic profile of myoblasts, particularly regulating oxidant metabolism and amino acid metabolism.

### 2.7. Trehalose Activated the Nrf2-NQO1 Axis and Promoted the Nuclear Translocation of Nrf2

To gain further insights into the protective mechanism of trehalose in myoblasts against oxidative stress, we investigated the impact of trehalose supplementation on the classic antioxidant system, the Keap1-Nrf2 pathway. Our results demonstrated that trehalose induced a significant upregulation of Nrf2 protein under oxidative stress conditions (Figure 7A). To further validate this observation, we utilized the specific inhibitor of Nrf2, ML385 (5 μM), and observed a decrease in myoblast viability and Nrf2 expression when exposed to H_2_O_2_ (Figure 7B–D). Notably, the beneficial effects of trehalose on myoblast viability and Nrf2 levels were significantly attenuated in the presence of ML385 (Figure 7B–D). Furthermore, we examined the expression of NQO1, a downstream protein of the Keap1-Nrf2 pathway, and observed a substantial increase in its expression with trehalose supplementation under oxidative stress, which was reversed when ML385 was present (Figure 7E). These findings strongly suggest that trehalose protects myoblasts from oxidative stress by activating the Nrf2-NQO1 axis.

To further explore the effect of trehalose supplementation on the localization of Nrf2, we investigated the expressions of Nrf2 in the nucleus and cytoplasm. Surprisingly, both trehalose treatment and pretreatment significantly promoted the nuclear translocation of Nrf2 (Figure 8A, B), indicating the activation of the Nrf2-NQO1 axis by trehalose. Previous studies have revealed two distinct mechanisms for the activation of the Keap1-Nrf2 pathway: canonical and non-canonical. The canonical mechanism is linked to a conformational change of Keap1, which leads to Nrf2 being released from the Keap1-Nrf2 complex and entering the nucleus to initiate the transcription of antioxidants. The non-canonical mechanism involves Keap1 binding to the p62 protein, which then facilitates the entry of Nrf2 into the nucleus. 

To elucidate the mechanism by which trehalose activates the Nrf2-NQO1 axis, we assessed the expression levels of the p62 protein before and after trehalose supplementation. Strikingly, both trehalose treatment and pretreatment significantly increased p62 expressions in cells under oxidative stress (Figure 8C,D). These results highlight the establishment of a positive feedback loop involving p62-Keap1-Nrf2-NQO1 in cells following both trehalose treatment and pretreatment. Trehalose treatment exhibits a therapeutic effect against short-term oxidative stress, whereas trehalose pretreatment demonstrates a preventive effect against prolonged oxidative stress.

## 3. Discussion

Oxidative stress in skeletal muscle has emerged as a major concern due to its crucial role as the largest organ in the human body, essential for vital functions and metabolism [35,36,37]. Trehalose, a natural non-toxic disaccharide, has been reported to have promising antioxidant properties in previous studies [38,39,40]. It has been previously demonstrated that trehalose can alleviate oxidative stress in liver cells [41], peripheral blood mononuclear cells [42], the spleen [43], the kidney [21], and so on. However, the specific role of trehalose in alleviating oxidative stress in skeletal muscle and the underlying mechanisms remain poorly understood. In this study, we used two interventional approaches to investigate the therapeutic and preventive effects of trehalose on myoblasts subjected to oxidative stress. Through comprehensive analyses of cell phenotype, cellular metabolism, and protein expression, we aimed to elucidate the metabolic mechanisms by which trehalose protects myoblasts against oxidative stress, thus shedding light on the remarkable antioxidant capacity of trehalose. 

### 3.1. Trehalose Regulates Oxidant Metabolism and Facilitates ROS Scavenging

As expected, our results showed that both short-term and prolonged exposure to H_2_O_2_ resulted in the abnormal accumulation of ROS in myoblasts, accompanied by an increase in cellular MDA levels, indicative of lipid peroxidation. Interestingly, the levels of GSH, a crucial component in ROS scavenging [44], showed different patterns of changes under short-term and long-term oxidative stress. Specifically, we observed a significant upregulation of GSH levels in myoblasts following short-term exposure (2 h, Appendix A), whereas prolonged exposure to H_2_O_2_ (12 + 24 h, Appendix A) resulted in a significant downregulation of GSH levels. Interestingly, a similar trend was observed for taurine, another antioxidant known to promote ROS scavenging [45,46]. This disparity in GSH and taurine levels could be attributed to the initiation of the antioxidant mechanism in myoblasts during short-term oxidative stress, whereas long-term oxidative stress is likely to have depleted the cellular pool of antioxidants.

Significantly, our results demonstrate that trehalose supplementation effectively promotes ROS scavenging and mitigates lipid peroxidation. In the therapeutic effect experiment, the HT cells exhibited a notable decrease in GSH and taurine levels compared to the H cells, approaching levels similar to those of the untreated controls (the C cells). This observation suggests that trehalose assists myoblasts in recovering from the adverse stress state, potentially involving the utilization of GSH and taurine for ROS scavenging, as evidenced by the significant reduction in glutamine and threonine levels. Interestingly, in the preventive effect experiment, there were no significant differences in GSH and taurine levels between the pTH and pH cells, both of which exhibited lower levels compared to the pC cells. Furthermore, the levels of glutamate and glycine, which are precursors of GSH, were significantly reduced following trehalose pretreatment. These intriguing results can be explained by the possibility that prolonged oxidative stress depletes GSH levels to such an extent that they have not yet returned to normal, leading to the utilization of available precursors for GSH synthesis. 

In summary, both trehalose treatment and pretreatment significantly impact glutathione metabolism and D-glutamine and D-glutamate metabolism in C2C12 cells exposed to H_2_O_2_. Our findings highlight the crucial role of trehalose in regulating oxidant metabolism and enhancing ROS scavenging in cells subjected to both short-term and prolonged oxidative stress. 

### 3.2. Trehalose Regulates Amino Acid Metabolism and the TCA Cycle

We observed that short-term exposure to H_2_O_2_ significantly increased glucose levels and decreased lactate levels, indicating an upregulation of glycolysis in myoblasts. Similarly, prolonged exposure to H_2_O_2_ significantly decreased lactate levels, while glucose levels remained unchanged. These results demonstrate that oxidative stress impairs the energy metabolism of myoblasts, consistent with previous studies in astrocytes [47]. These alterations in glucose and lactate levels suggest a disruption of the energy metabolism in myoblasts under oxidative stress conditions.

Creatine, a key player in interconverting with phosphocreatine and storing ATP-derived energy, exhibited a significant decrease in levels following prolonged oxidative stress, while phosphocreatine levels remained relatively stable. Notably, no significant changes in creatine and phosphocreatine levels were observed after short-term oxidative stress. This discrepancy may be attributed to the increased energy demand during prolonged oxidative stress, resulting in a more pronounced disruption of energy metabolism in myoblasts. Interestingly, we observed a significant increase in myo-inositol levels accompanied by a notable decrease in amino acids such as isoleucine, leucine, valine, tryptophan, and tyrosine following trehalose pretreatment. These amino acids are likely utilized to enter the TCA cycle for energy production. These results suggest that trehalose pretreatment may assist myoblasts in combating oxidative stress by upregulating the compensatory pathway of the TCA cycle, especially under conditions of energy deprivation induced by prolonged oxidative stress.

### 3.3. Trehalose Increases the Nuclear Localization of Nrf2 by Increasing the Levels of P62 and Nrf2

Previous research has elucidated the crucial involvement of various signaling pathways in oxidative stress, including the Keap1-Nrf2 pathway [13], the nuclear factor κB (NF-κB) pathway [48], and the mitogen-activated protein kinase (MAPK) pathway [42,49]. Among them, the Keap1-Nrf2 system stands out as a vital and evolutionarily conserved defense mechanism in cells to combat oxidative stress [50,51]. In our study, we observed that trehalose exerts a positive regulatory effect on Nrf2 and activates the Nrf2-NQO1 axis under oxidative stress conditions, as demonstrated by the significant decrease in cell viability and NQO1 expression upon ML385 intervention. Importantly, the upregulation of p62 expression and the enhanced nuclear translocation of Nrf2 following trehalose treatment or pretreatment indicate the activation of the Keap1-Nrf2 pathway through a non-canonical mechanism. 

Although our results are consistent with previous studies demonstrating the capacity of trehalose to activate the Keap1-Nrf2 pathway to combat oxidative stress [52,53,54], there are still some unresolved questions as some reports have suggested that trehalose can suppress the Keap1-Nrf2 pathway to safeguard against spleen damage [43]. Furthermore, the involvement of autophagy in the antioxidant effect of trehalose remains a subject of debate, as some studies suggest its promotion [41,55], while others propose that the antioxidant effect of trehalose is independent of autophagy [21,25,56]. Further research using alternative methods is needed to gain insight into how trehalose reduces oxidative stress and to understand the interplay between its various effects.

In addition, in this study, trehalose was added to treat cells either after completion of the H_2_O_2_ modelling process, with a treatment time of 24 h, or during the entire process, with a pretreatment time of 48 h. Trehalose pretreatment showed a strong protective effect after the third 24 h supplementation (Appendix A). Certainly, it is a valuable attempt to investigate the temporal effects of trehalose treatment over longer time points than the third 24 h supplementation to determine whether the protective effects persist or diminish over time. We will be doing this in the future.

In conclusion, our study has revealed the remarkable antioxidant capacity of trehalose in skeletal muscle cells and elucidated the underlying molecular mechanisms by which it defends against oxidative stress through the activation of the Nrf2-NQO1 axis. Both trehalose treatment and pretreatment significantly enhance the expression levels of P62 and Nrf2, leading to increased nuclear localization of Nrf2 and upregulation of NQO1, a critical antioxidant protein. Importantly, this is the first investigation to explore the protective effects of trehalose on C2C12 myoblasts under oxidative stress. Our results provide valuable insights into the potential of trehalose supplementation as a promising and effective strategy for alleviating oxidative stress in skeletal muscle. This study lays the foundation for further research and holds promise for the development of innovative therapeutic approaches targeting oxidative stress-related disorders in skeletal muscle.

## 4. Materials and Methods

### 4.1. Reagents

D- (+)-Trehalose dihydrate and the total antioxidant capacity (T-AOC) assay kit were purchased from Beyotime Biotechnology (Shanghai, China). ML385 and the MDA assay kit were purchased from Solarbio (Beijing, China). The primary antibodies used were Nrf2 (16396-1-AP, Proteintech, Wuhan, China), NQO1 (AC58849, Acmec, Wuhan China), CAT (16396-1-AP, Proteintech, Wuhan, China), p62 (18420-I-AP, Proteintech, Wuhan, China), Histone3 (CJ36131, Bioworld, Nanjing, China), and GAPDH (G0100, Lablead, Xiamen, China).

### 4.2. Cell Culture

Murine C2C12 myoblasts were procured from the American-type culture collection (ATCC) (IMMOCELL, Xiaman, China). They were grown in a humidified atmosphere (37 °C, 5% CO_2_) with DMEM medium (HyClone, Logan, UT, USA) supplemented with 10% (*v*/*v*) fetal bovine serum (FBS, Gibco, Gaithersburg, MD, USA), 100 U/mL penicillin, and 100 mg/mL streptomycin. The cell experimental design is depicted in Appendix A. Cell morphological images were captured using a fluorescence microscope (Motic, AE31E, Xiamen, China).

Four groups of C2C12 cells (C, T, H, HT) were used to evaluate the therapeutic effect of trehalose treatment on cells with short-term exposure to H_2_O_2_. Furthermore, three groups of C2C12 cells (pC, pH, pTH) were used to assess the preventive effect of trehalose pretreatment on cells with prolonged exposure to H_2_O_2_. The final concentrations of H_2_O_2_ and trehalose were 200 μM and 10 mM, respectively.

### 4.3. Cell Viability Measurements

Cell viability was assayed by a Cell Titer 96 Aqueous Solution Cell Proliferation Assay Kit (Promega, Madison, WI, USA). Myoblasts were seeded at a density of 5 × 10^3^ (200 μL per well) in 96-well plates and cultured as described above, with five replicates for each group of samples (*n* = 5). Then, 20 μL of MTS (3-(4, 5-dimethylthiazol-2-yl)-5-(3-carboxymethoxyphenyl)-2-(4-sulfophenyl)2H-tetrazolium) was added to each well, and the cells were incubated in the dark at 37 °C for 3 h followed by measuring the absorbance of formazan at a wavelength of 490 nm on a multimode microplate reader (Biotech, Winooski, VT, USA).

### 4.4. Intracellular ROS Measurements

The intracellular ROS levels were measured using a membrane-permeable probe, H_2_DCFDA (2′, 7′-dichlorofluorescein diacetate, Sigma-Aldrich, St. Louis, MO, USA). After trypsin digestion and washing the myoblasts with PBS, 10 μM H_2_DCFDA was added to the cell precipitates and incubated at 37 °C for 40 min. The cells were then washed with PBS three times and transferred to a black 96-well plate to measure the fluorescence intensity at 525 nm (*n* = 4). Fluorescence images were obtained using a confocal microscope (Motic, AE31E, Xiamen, China).

### 4.5. Lipid Peroxidation Assessed Using the MDA Assay

After cell lysis, the supernatant was collected, and the MDA assay kit (Solarbio, Beijing, China) was used according to the manufacturer’s instructions. The corresponding reagent was added to the sample tubes and control tubes. The samples were boiled at 100 °C for 1 h and then centrifuged to obtain the supernatant. The supernatant was transferred to a 96-well plate, and the absorbance at 532 nm and 600 nm was measured (*n* = 4). The difference in absorbance between the two wavelengths was used to calculate the MDA content.

### 4.6. Assay of Cellular T-AOC

The total antioxidant capacity of cells was measured using the ABTS (2,2-diazo-di(3-ethyl-benzothiazol-6-sulfonic acid) diammonium salt) method. After cell lysis, the supernatant was obtained and diluted. The diluted supernatant was then added to the pre-prepared working reagent following the instructions provided in the T-AOC assay kit (Beyotime, Shanghai, China). The absorbance at 734 nm was measured (*n* = 4), and the total antioxidant capacity of the sample was calculated based on a standard curve.

### 4.7. Extraction of Cellular Metabolites

Aqueous metabolites were extracted from myoblasts using the dual-phase extraction method as previously described [57]. Myoblasts (approximately 5 × 10^6^ per dish) were harvested after being washed three times with prechilled PBS. To extract intracellular metabolites, a mixture of methanol, chloroform, and water in a volume ratio of 4:4:2.85 was added. The polar cell extracts were then lyophilized using a Freeze dryer (LGJ-10E, Foring Technology Development, Beijing Co., Ltd., Beijing, China) below −65 °C. The lyophilized extracts were suspended in 550 μL of NMR buffer (100% D_2_O, 0.05 mM TSP, pH 7.4). Sodium 3-(trimethylsilyl) propionate-2,2,3,3-d4 (TSP) was added as an internal standard to calibrate the chemical shifts (δ 0.00) in NMR spectra and quantitatively measure the levels of the metabolites. All re-dissolved samples were transferred into 5 mm NMR tubes and centrifuged prior to the subsequent NMR experiments. In this study, 10 samples from each of the experimental groups were used for the cell metabolomics analysis.

### 4.8. Acquisition of NMR Spectra 

NMR spectra were acquired at 298 K using a Bruker Avance III 850 MHz spectrometer (Bruker BioSpin, Ettlingen, Germany) equipped with a TCI cryoprobe. One-dimensional (1D) ^1^H spectra were obtained using the NOESYGPPR1D pulse sequence [RD-G1-90°-t_1_-90°-τ_m_-G2-90°-ACQ]. During the relaxation delay (t_1_ = 10 μs) and mixing time (τ_m_ = 10 ms), pulsed gradients G1 and G2 were applied to enhance water suppression. A total of 64 transients were collected into 64 K data points, with a spectral width of 20 ppm and an acquisition time (ACQ) of 2.66 s. An additional relaxation delay of 4 s was included. 

For the two-dimensional (2D) ^1^H–^13^C heteronuclear single quantum coherence (HSQC) spectrum, a data matrix of 1024 × 256 points was used, along with a relaxation delay of 1.5 s. The 2D total correlation spectroscopy (TOCSY) spectrum was recorded with a data matrix of 2048 × 256 points and a relaxation delay of 1.5 s.

### 4.9. NMR Data Preprocessing and Resonance Assignments

Free induction delay (FID) signals were processed by applying an exponential function with a line-broadening factor of 0.3 Hz before Fourier transformation. The MestReNova software (version 9.0 Mestrelab Research S. L, Santiago, Spain) was used for phase adjustment, baseline correction and calibration of each 1D ^1^H-NMR spectrum. The regions of *δ* 0.60–9.50 ppm were divided into 0.001 ppm units, excluding the water resonance (δ 4.70–4.90 ppm) and trehalose resonances (δ 3.44–3.48, 3.63–3.66, 3.75–3.79, 3.81–3.89, 5.21–5.23 ppm). The spectral peak integrals were normalized by the sum of the total spectral peak integrals to ensure accurate comparison between samples.

For preliminary identification of metabolites, the peaks in the 1D ^1^H-NMR spectra were initially fitted using Chenomx NMR Suite (Version 8.5, Chenomx Inc., Edmonton, AB, Canada). Additionally, the identification process was aided by referring to the Human Metabolome Database (HMDB) (http://www.hmdb.ca/, accessed on 22 November 2022)) and relevant studies [58,59]. To further confirm the identification results, 2D ^1^H-^13^C heteronuclear single quantum coherence (HSQC) spectra and 2D ^1^H-^1^H total correlation spectroscopy (TOCSY) spectra were utilized.

The relative intensity of each identified metabolite was calculated based on the relative integral of singlet or nonoverlapped peaks in each NMR spectrum. These relative intensities were used to represent the relative levels of the metabolites. The data were expressed as mean ± standard deviation (SD), providing a quantitative assessment of the metabolite levels.

### 4.10. Multivariate Analysis and Univariate Analysis of NMR Data

The normalized 1D ^1^H-NMR spectral data were subjected to multivariate statistical analysis using the SIMCA software (Version 14.1, MKS Umetrics AB, Umea, Sweden). Prior to analysis, the data were processed with univariate scaling to ensure comparability among the variables. Initially, unsupervised principal component analysis (PCA) was performed to visualize the overall metabolic patterns and assess the clustering between samples.

Subsequently, supervised analysis methods, namely partial least squares discriminant analysis (PLS-DA) and orthogonal partial least squares discriminant analysis (OPLS-DA), were used to achieve improved differentiation of the metabolic patterns based on the distinct tendencies observed between sample groups. To evaluate the reliability and robustness of the PLS-DA and OPLS-DA models, a cross-validation procedure was performed using a response permutation test (RPT) with 200 cycles.

To identify the metabolites contributing significantly to the metabolic separation between groups, the variable influence on projection (VIP) values were calculated based on the OPLS-DA model. Significant metabolites were identified using a criterion of VIP ≥ 1, which were further analyzed for their potential biological relevance. 

To quantitatively compare the relative levels of metabolites among different groups, an independent sample *t*-test analysis was performed using the SPSS software (Version 23, IBM, Armon, NY, USA). Differential metabolites were identified using a criterion of *p* < 0.05. 

In addition, characteristic metabolites were identified using two criteria of *p* < 0.05 and VIP ≥ 1, which exhibited both statistical significance and biological relevance.

### 4.11. Metabolic Pathway Analysis

Significantly altered metabolic pathways were identified using the MetaboAnalyst 5.0 web server (https://www.metaboanalyst.ca, accessed on 10 March 2023) in combination with the KEGG database (Kyoto Encyclopedia of Genes and Genomes). Pathway enrichment analysis was conducted to assess the significance of metabolite enrichment within specific pathways based on the concentrations of all identified metabolites. A *p*-value threshold of *p* < 0.05 was used to determine significant enrichment. Furthermore, pathway topological analysis was performed to evaluate the importance of the identified pathways. The pathway impact value (PIV) was utilized as a measure of pathway significance, with higher PIV values indicating greater pathway importance. In this study, a PIV threshold of >0.2 was used to identify metabolic pathways with substantial changes based on pairwise comparisons. Significantly altered metabolic pathways were identified using two criteria of *p* < 0.05 and PIV > 0.2.

### 4.12. Inhibition of Nrf2 Activity

To inhibit the Nrf2 activity, the Nrf2 inhibitor ML385 was prepared as a stock solution in dimethyl sulfoxide (DMSO) and adjusted to a final concentration of 5 μM in the medium. The concentration of ML385 for treating the C2C12 myoblasts was determined primarily based on those values reported in the literature [60,61,62]. Prior to the intervention experiments, myoblasts were incubated with the ML385-containing medium for 12 h. Subsequently, the cells were cultured and treated according to the intervention methods described in Appendix A.

### 4.13. Extraction of Nuclear and Cytoplasmic Proteins

After washing the myoblasts with PBS, they were harvested by centrifugation to remove the supernatants. The extraction of proteins from the cell nucleus and cytoplasm was performed using the nuclear and cytoplasmic protein extraction kit (Beyotime, Shanghai, China), following the manufacturer’s instructions.

### 4.14. Measurements of Protein Expressions

Intracellular proteins were extracted using RIPA lysis buffer (Solarbio Technology Co., Ltd., Beijing, China) supplemented with protease inhibitor and phosphorylation protease inhibitor cocktails (Thermo Fisher, Waltham, MA, USA). The mixture was sonicated for 35 s and then centrifuged at 12,000× *g* for 10 min at 4 °C to collect the supernatant. The total protein concentration in the supernatant was determined using a BCA protein assay kit (LabLead, Xiamen, China), followed by the addition of an appropriate amount of SDS loading buffer. The denatured proteins were separated by SDS-PAGE (10–15% gel) through electrophoresis and transferred onto PVDF membranes (GE Healthcare, Shanghai, China). The membranes were then blocked with BSA solution (Beyotime Biotechnology, Shanghai, China) for 1 h. Subsequently, the membranes were incubated with specific primary antibodies and appropriate secondary antibodies. Protein bands of interest were visualized using an enhanced chemiluminescence reagent (ECL, Beyotime, Shanghai, China) and captured using ChemiScope Analysis software (ChemiScope 6000, CLiNX, Shanghai, China).

### 4.15. General Statistical Analysis

Experimental data were presented as mean ± SD. To assess the quantitative differences between groups, statistical analysis was performed using Student’s *t*-test with GraphPad Prism software (Version 8.0, La Jolla, San Diego, CA, USA). The statistical significance was as follows: *p* > 0.05 (ns, not significant), *p* < 0.05 (*), *p* < 0.01 (**), *p* < 0.001 (***), *p* < 0.0001 (****).

## Figures and Tables

**Figure 1 ijms-24-13346-f001:**
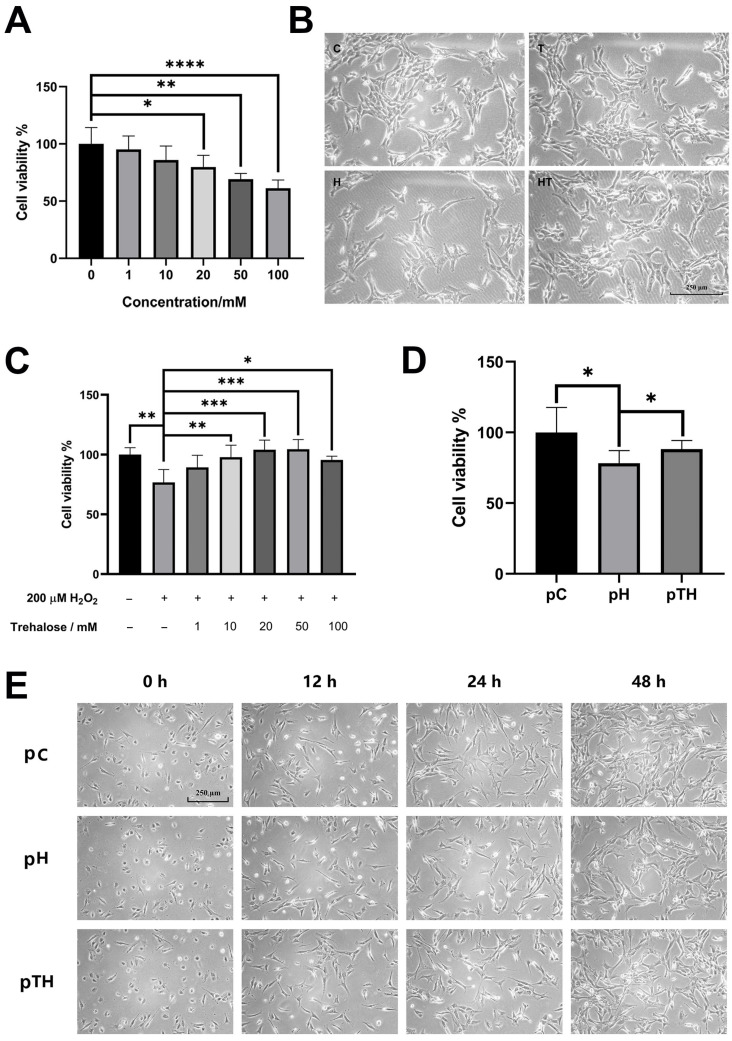
Trehalose treatment and pretreatment protected C2C12 cells against the H_2_O_2_-induced decrease in cell viability. (**A**) Cell viability after treatment with different concentrations of trehalose (*n* = 5). (**B**) Representative morphological images of the C, T, H, and HT cells. Scale bar, 100 μm. (**C**) Cell viability after treatment with H_2_O_2_ (200 μM) and trehalose at different concentrations (*n* = 5). (**D**) Cell viability after treatment with H_2_O_2_ (200 μM) and pretreatment with trehalose at 10 mM (*n* = 5). (**E**) Representative morphological images of the pC, pH, and pHT cells. Scale bar, 100 μm. The concentrations of H_2_O_2_ and trehalose were 200 μM and 10 mM, respectively. Statistical significance: *, *p* < 0.05; **, *p* < 0.01; ***, *p* < 0.001; ****, *p* < 0.0001.

**Figure 2 ijms-24-13346-f002:**
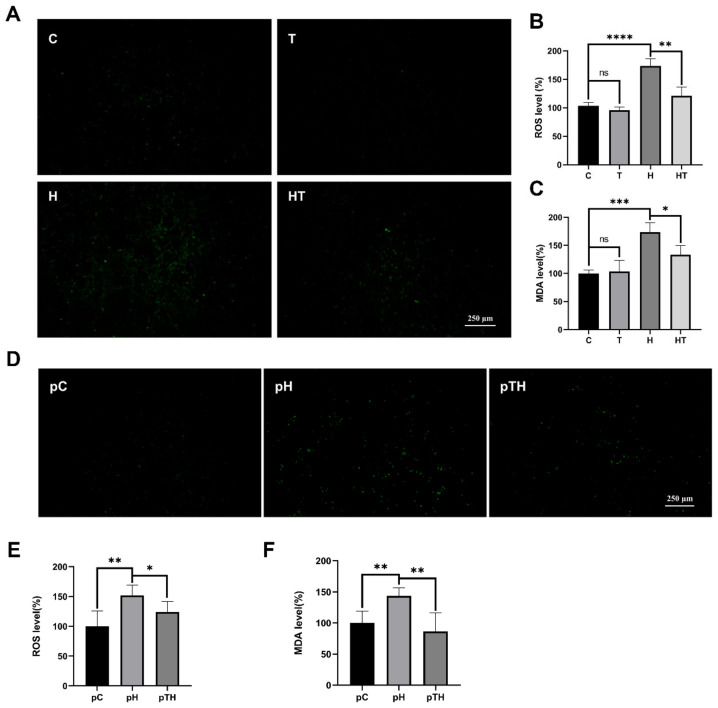
Trehalose treatment and pretreatment promoted the scavenging of H_2_O_2_-induced ROS in C2C12 cells. (**A**) Representative confocal microscopy image of H_2_DCFDA-stained C, T, H, and HT cells. Scale bar, 50 µm. (**B**) Intracellular ROS levels of the C, T, H, and HT cells detected by oxidized H_2_DCFDA fluorescence signals (*n* = 4). (**C**) Intracellular MDA levels of the C, T, H, and HT cells measured by the MDA assay (*n* = 4). (**D**) Representative confocal microscopy image of the pC, pH, and pHT cells. Scale bar, 50 µm. (**E**) Intracellular ROS levels of the pC, pH, and pHT cells (*n* = 4). (**F**) Intracellular MDA levels of the pC, pH, and pHT cells (*n* = 3). *p* > 0.05, ns; *p* < 0.05, *; *p* < 0.01, **; *p* < 0.001, ***; *p* < 0.0001, ****.

**Figure 3 ijms-24-13346-f003:**
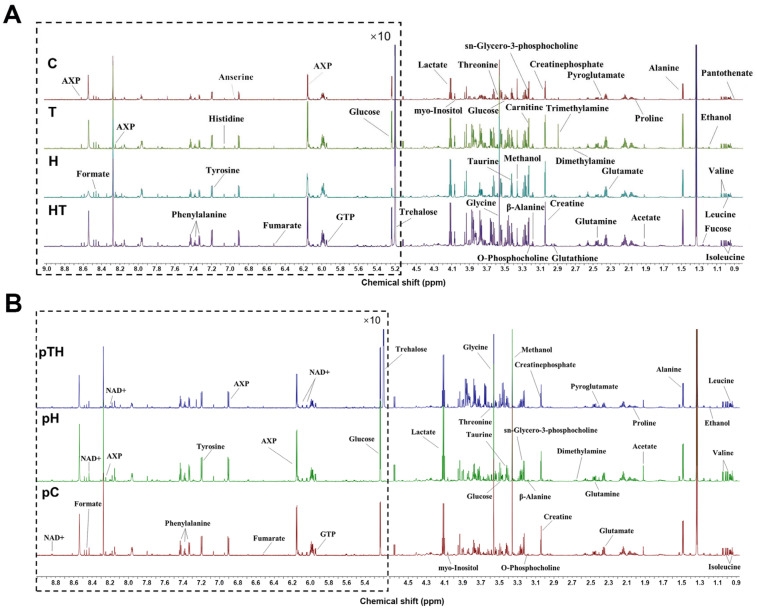
Typical 1D ^1^H-NMR spectra of C2C12 cells after treatment with H_2_O_2_ and trehalose. (**A**) NMR spectra of aqueous metabolites from the C, T, H, and HT cells. (**B**) NMR spectra of aqueous metabolites from the pC, pH, and pTH cells. In all the NMR spectra, the vertical scale remained constant, and the water peak signal was removed (4.70–4.90 ppm). Ten times magnification was performed in the 4.90–9.00 ppm region for the clarity. Abbreviations: GTP, guanosine triphosphate; AXP, adenine mono/di/triphosphate.

**Figure 4 ijms-24-13346-f004:**
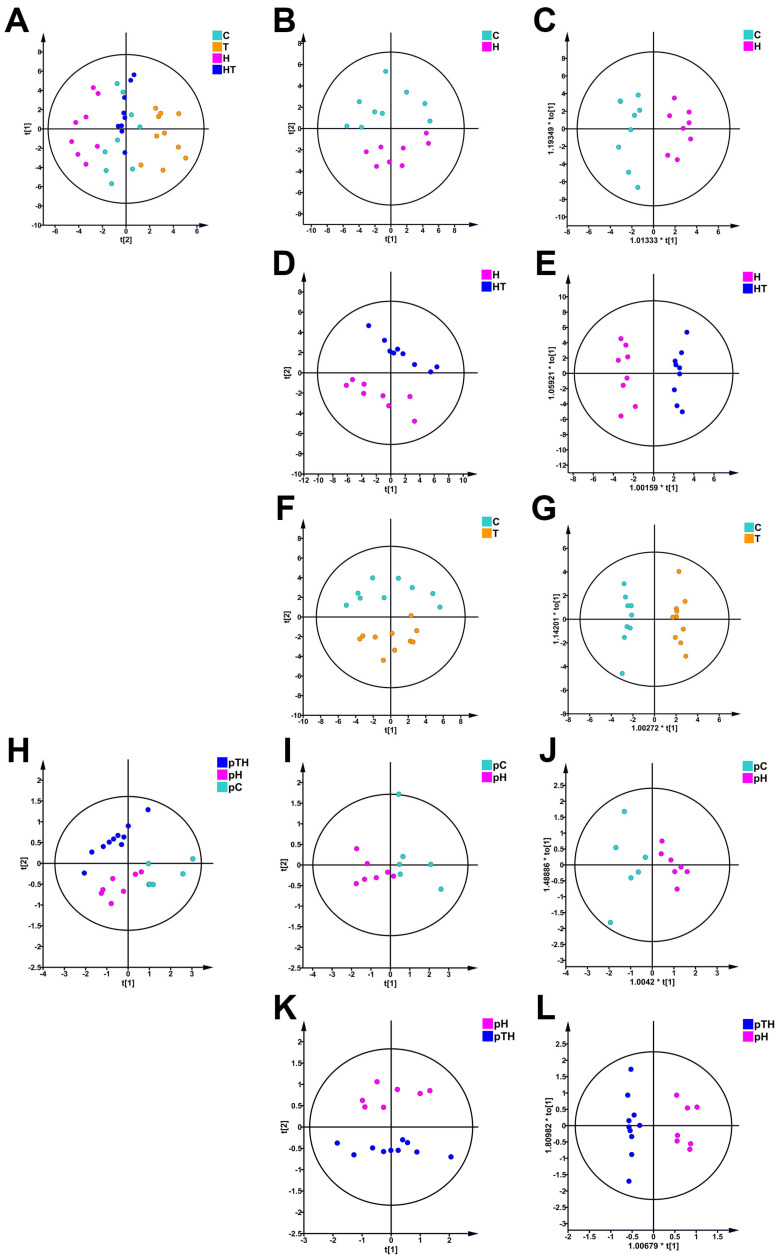
Multivariate statistical analyses for 1D ^1^H-NMR spectral data of C2C12 cells after treatment with H_2_O_2_ and trehalose. (**A**) PCA scores plot of the C, H, T, and HT groups for the trehalose treatment experiment. (**B**,**D**,**F**) PCA scores plots for H vs. C, HT vs. H, and T vs. C. (**C**,**E**,**G**) OPLS-DA scores plots for H vs. C, HT vs. H, and T vs. C. (**H**) PCA scores plot of the pC, pH, and pTH groups for the trehalose pretreatment experiment. (**I**,**K**) PCA scores plots for pH vs. pC and pTH vs. pH. (**J**,**L**) OPLS-DA scores plots for pH vs. pC and pTH vs. pH.

**Figure 5 ijms-24-13346-f005:**
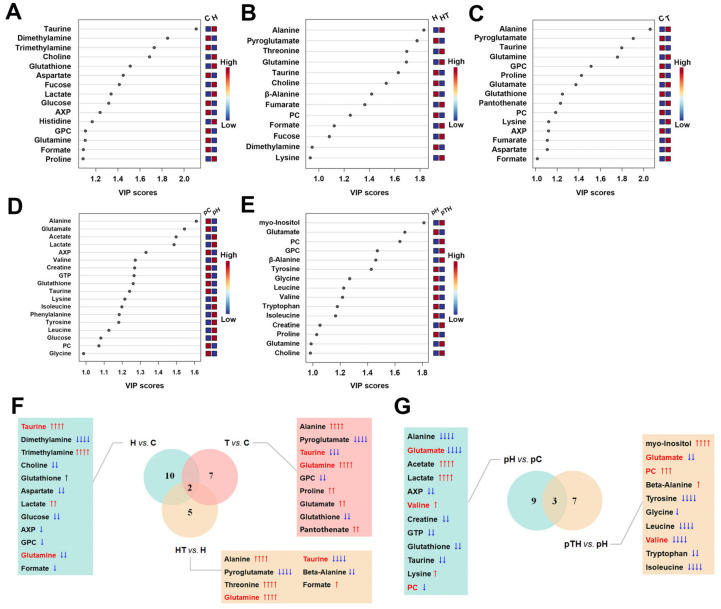
Identification of significant and characteristic metabolites from pairwise comparisons between groups of C2C12 cells. (**A**–**E**) VIP score-ranking plots of significant metabolites identified from the OPLS-DA model for (**A**) H vs. C, (**B**) HT vs. H, (**C**) T vs. C, (**D)** pH vs. pC, and (**E**) pTH vs. pH. (**F**) Venn diagrams of characteristic metabolites identified from pairwise comparisons of H vs. C, T vs. C and HT vs. H. (**G**) Venn diagrams of characteristic metabolites identified from pairwise comparisons of pH vs. PC and pTH vs. pH. Metabolites in red were characteristic metabolites shared by the (**F**) three/(**G**) two pairwise comparisons. These metabolites were ranked in descending order according to the VIP scores (Appendix A). Statistical significance: ↓/↑, *p* < 0.05; ↓↓/↑↑, *p* < 0.01; ↓↓↓/↑↑↑, *p* < 0.001; ↓↓↓↓/↑↑↑↑, *p* < 0.0001.

**Figure 6 ijms-24-13346-f006:**
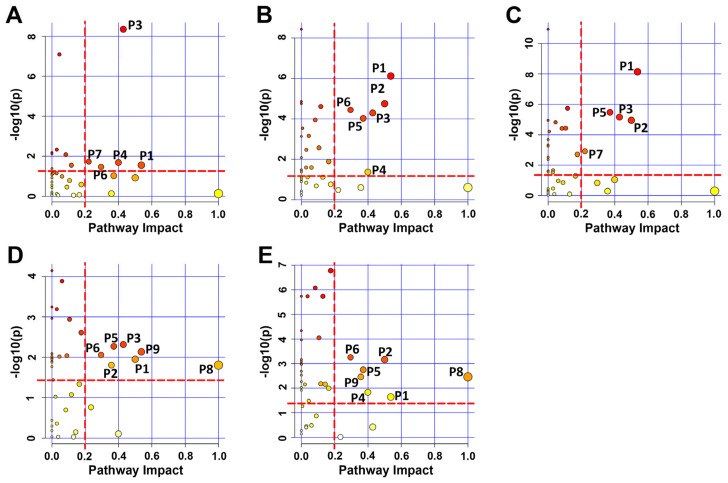
Identification of significantly altered metabolic pathways from pairwise comparisons between groups of C2C12 cells. (**A**) H vs. C, (**B**) HT vs. H, (**C**) T vs. C, (**D**) pH vs. pC, and (**E**) pTH vs. pH. Significantly altered metabolic pathways were identified with two criteria of pathway impact values > 0.2 and *p* values < 0.05 using the pathway analysis module provided by MetaboAnalyst 5.0 webserver. The characters in the figure represent the following: P1: alanine, aspartate, and glutamate metabolism; P2: D-Glutamine and D-glutamate metabolism; P3: taurine and hypotaurine metabolism; P4: β-alanine metabolism; P5: glutathione metabolism; P6: glycine, serine, and threonine metabolism; P7: histidine metabolism; P8: phenylalanine, tyrosine, and tryptophan biosynthesis; P9: phenylalanine metabolism.

**Figure 7 ijms-24-13346-f007:**
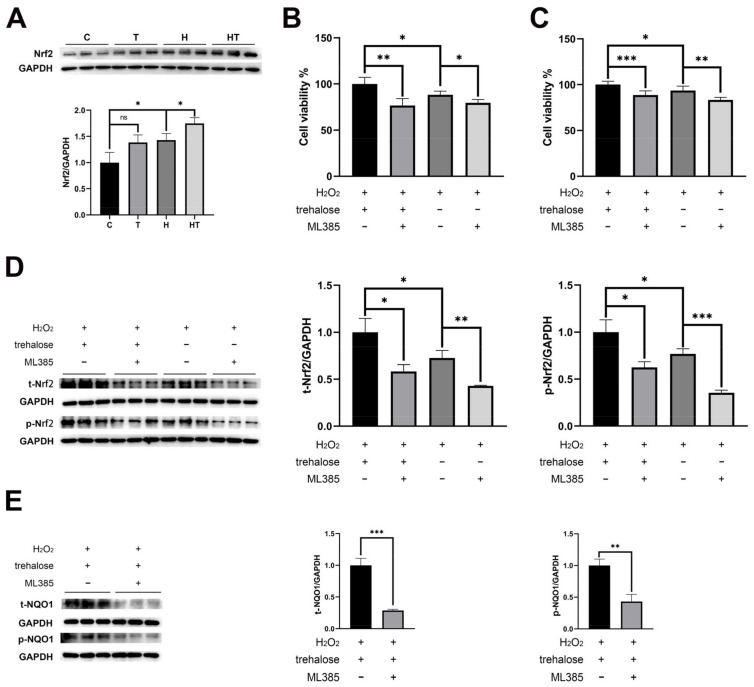
Trehalose treatment and pretreatment activated the Keap1-Nrf2 pathway. (**A**) Expression levels of Nrf2 in the C, T, H, and HT cells (*n* = 3). (**B**) Cell viability in the trehalose treatment experiment under short-term oxidative stress (*n* = 5). (**C**) Cell viability in the trehalose pretreatment experiment under prolonged oxidative stress (*n* = 5). (**D**) Expression levels of Nrf2 in the trehalose treatment experiment (t-Nrf2) and the trehalose pretreatment experiment (p-Nrf2) (*n* = 3). (**E**) Expression levels of NQO1 in the trehalose treatment experiment (t-NQO1) and the trehalose pretreatment experiment (p-NQO1) (*n* = 3). *p* < 0.05, *; *p* < 0.01, **; *p* < 0.001, ***.

**Figure 8 ijms-24-13346-f008:**
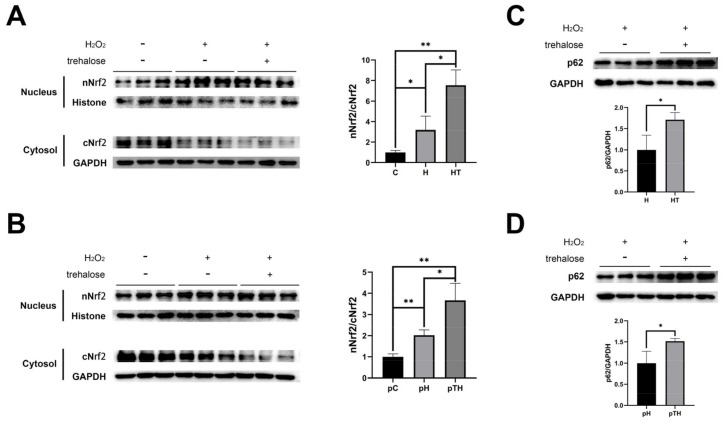
Trehalose treatment and pretreatment enhanced the expression levels of p62 and Nrf2 in cell nuclei. (**A**,**B**) Expression levels of Nrf2 in cell nuclei relative to cytoplasm in the trehalose treatment experiment (**A**) (*n* = 3) and the trehalose pretreatment experiment (**B**) (*n* = 3). (**C**,**D**) Expression levels of p62 in the pC, pH, and pTH cells (**C**) (*n* = 3) and the pH and pTH cells (**D**) (*n* = 3). Statistical significance: *p* < 0.05, *; *p* < 0.01, **.

**Table 1 ijms-24-13346-t001:** Univariate statistical analysis of relative integrals of metabolites in the C, T, H, and HT groups of C2C12 cells.

Metabolite	Mean ± SD				*t*-Test		
	C	T	H	HT	H vs. C	T vs. C	HT vs. H
Pantothenate	0.038 ± 0.004	0.045 ± 0.005	0.039 ± 0.006	0.042 ± 0.003	ns	**	ns
Leucine	0.487 ± 0.023	0.496 ± 0.023	0.488 ± 0.021	0.488 ± 0.018	ns	ns	ns
Isoleucine	0.244 ± 0.016	0.245 ± 0.010	0.247 ± 0.014	0.248 ± 0.012	ns	ns	ns
Valine	0.249 ± 0.017	0.245 ± 0.011	0.252 ± 0.012	0.249 ± 0.012	ns	ns	ns
Fucose	0.199 ± 0.017	0.198 ± 0.015	0.213 ± 0.015	0.221 ± 0.011	ns	ns	ns
Alanine	1.411 ± 0.069	1.654 ± 0.050	1.330 ± 0.046	1.563 ± 0.060	*	****	****
Acetate	0.213 ± 0.068	0.237 ± 0.032	0.196 ± 0.033	0.210 ± 0.038	ns	ns	ns
Proline	0.109 ± 0.004	0.102 ± 0.005	0.109 ± 0.005	0.104 ± 0.005	ns	**	ns
Glutamate	0.384 ± 0.024	0.414 ± 0.019	0.373 ± 0.017	0.376 ± 0.022	ns	**	ns
Pyroglutamate	0.116 ± 0.009	0.077 ± 0.005	0.116 ± 0.008	0.080 ± 0.004	ns	****	****
Glutamine	0.189 ± 0.014	0.221 ± 0.015	0.168 ± 0.014	0.206 ± 0.008	**	****	****
Dimethylamine	0.022 ± 0.005	0.029 ± 0.022	0.009 ± 0.005	0.004 ± 0.000	****	ns	ns
Aspartate	0.041 ± 0.003	0.043 ± 0.004	0.032 ± 0.007	0.029 ± 0.003	**	ns	ns
Trimethylamine	0.170 ± 0.060	0.306 ± 0.337	0.045 ± 0.053	0.014 ± 0.004	****	ns	
Glutathione	0.110 ± 0.014	0.093 ± 0.010	0.125 ± 0.012	0.113 ± 0.009	*	**	*
Lysine	0.106 ± 0.009	0.115 ± 0.008	0.113 ± 0.012	0.123 ± 0.009	ns	*	ns
Creatine	1.568 ± 0.139	1.621 ± 0.140	1.547 ± 0.143	1.551 ± 0.073	ns	ns	ns
Phosphocreatine	1.216 ± 0.173	1.133 ± 0.176	1.107 ± 0.150	0.992 ± 0.099	ns	ns	ns
Beta-Alanine	0.205 ± 0.024	0.198 ± 0.017	0.200 ± 0.016	0.172 ± 0.012	ns	ns	**
Choline	0.083 ± 0.022	0.077 ± 0.013	0.152 ± 0.042	0.076 ± 0.009	**	ns	****
PC	1.976 ± 0.139	1.812 ± 0.105	1.950 ± 0.189	1.642 ± 0.149	ns	**	**
GPC	1.484 ± 0.169	1.281 ± 0.066	1.268 ± 0.189	1.132 ± 0.059	*	**	ns
Taurine	0.438 ± 0.023	0.398 ± 0.017	0.481 ± 0.013	0.433 ± 0.024	****	***	****
Glucose	0.395 ± 0.057	0.365 ± 0.052	0.276 ± 0.085	0.318 ± 0.056	**	ns	ns
Glycine	2.996 ± 0.165	2.896 ± 0.117	2.909 ± 0.215	2.746 ± 0.117	ns	ns	ns
Threonine	0.183 ± 0.023	0.196 ± 0.016	0.174 ± 0.004	0.186 ± 0.004	ns	ns	****
Myo-inositol	1.343 ± 0.140	1.341 ± 0.078	1.287 ± 0.092	1.305 ± 0.075	ns	ns	ns
Lactate	1.447 ± 0.102	1.429 ± 0.071	1.595 ± 0.093	1.602 ± 0.072	**	ns	ns
GTP	0.103 ± 0.008	0.099 ± 0.007	0.097 ± 0.006	0.091 ± 0.006	ns	ns	ns
Fumarate	0.008 ± 0.002	0.010 ± 0.001	0.008 ± 0.002	0.009 ± 0.001	ns	*	*
Histidine	0.023 ± 0.002	0.022 ± 0.002	0.025 ± 0.002	0.024 ± 0.001	ns	ns	ns
Tyrosine	0.169 ± 0.010	0.162 ± 0.009	0.165 ± 0.006	0.162 ± 0.005	ns	ns	ns
Phenylalanine	0.088 ± 0.007	0.087 ± 0.004	0.086 ± 0.005	0.088 ± 0.004	ns	ns	ns
AXP	0.456 ± 0.031	0.437 ± 0.019	0.417 ± 0.028	0.400 ± 0.028	*	ns	ns
Formate	0.021 ± 0.003	0.024 ± 0.005	0.018 ± 0.001	0.020 ± 0.002	*	ns	*

Note: ns *p* > 0.05, * *p* < 0.05, ** *p* < 0.01, *** *p* < 0.001, **** *p* < 0.0001. Red/blue indicates an increase/decrease in the relative metabolite levels, respectively.

**Table 2 ijms-24-13346-t002:** Univariate statistical analysis of relative integrals of metabolites in the pC, pH, and pTH groups of C2C12 cells.

Metabolite	Mean ± SD			*t*-Test	
	pC	pH	pTH	pH vs. pC	pTH vs. pH
Leucine	0.338 ± 0.008	0.343 ± 0.011	0.316 ± 0.011	ns	****
Isoleucine	0.326 ± 0.009	0.336 ± 0.011	0.310 ± 0.011	ns	****
Valine	0.325 ± 0.007	0.337 ± 0.011	0.310 ± 0.009	*	****
Alanine	3.369 ± 0.142	2.929 ± 0.160	2.905 ± 0.231	****	ns
Acetate	0.397 ± 0.027	0.492 ± 0.032	0.456 ± 0.030	****	*
Proline	0.181 ± 0.017	0.165 ± 0.012	0.157 ± 0.012	ns	ns
Glutamate	2.126 ± 0.094	1.819 ± 0.116	1.625 ± 0.103	****	**
Pyroglutamate	0.218 ± 0.027	0.221 ± 0.013	0.210 ± 0.009	ns	ns
Glutamine	1.395 ± 0.101	1.352 ± 0.079	1.418 ± 0.086	ns	ns
Dimethylamine	0.015 ± 0.003	0.014 ± 0.003	0.011 ± 0.003	ns	ns
Aspartate	0.021 ± 0.008	0.021 ± 0.008	0.020 ± 0.007	ns	ns
Glutathione	0.266 ± 0.042	0.185 ± 0.044	0.169 ± 0.047	**	ns
lysine	0.143 ± 0.021	0.170 ± 0.014	0.164 ± 0.012	*	ns
Creatine	1.275 ± 0.139	1.027 ± 0.084	1.093 ± 0.086	**	ns
Phosphocreatine	0.806 ± 0.050	0.779 ± 0.069	0.855 ± 0.095	ns	ns
Beta-Alanine	0.030 ± 0.003	0.027 ± 0.003	0.031 ± 0.003	ns	*
Choline	0.047 ± 0.010	0.044 ± 0.008	0.052 ± 0.008	ns	ns
PC	1.302 ± 0.174	1.072 ± 0.094	1.271 ± 0.101	*	***
GPC	1.023 ± 0.150	0.918 ± 0.087	1.061 ± 0.106	ns	**
Taurine	0.563 ± 0.045	0.489 ± 0.029	0.481 ± 0.035	**	ns
Glucose	0.484 ± 0.122	0.607 ± 0.077	0.611 ± 0.092	ns	ns
Glycine	3.758 ± 0.439	3.215 ± 0.254	2.951 ± 0.247	*	*
Threonine	0.209 ± 0.007	0.190 ± 0.011	0.197 ± 0.010	**	ns
Myo-inositol	0.409 ± 0.058	0.384 ± 0.038	0.482 ± 0.034	ns	****
Lactate	5.156 ± 0.214	5.786 ± 0.164	5.812 ± 0.220	****	ns
GTP	0.103 ± 0.010	0.085 ± 0.011	0.079 ± 0.010	**	ns
AXP	0.584 ± 0.056	0.469 ± 0.055	0.429 ± 0.047	**	ns
Fumarate	0.005 ± 0.001	0.004 ± 0.001	0.004 ± 0.001	ns	ns
Tyrosine	0.268 ± 0.005	0.270 ± 0.005	0.250 ± 0.004	ns	****
Phenylalanine	0.206 ± 0.004	0.209 ± 0.007	0.217 ± 0.008	ns	*
Tryptophan	0.017 ± 0.001	0.017 ± 0.001	0.015 ± 0.001	ns	**
NAD+	0.035 ± 0.005	0.029 ± 0.004	0.030 ± 0.003	*	ns
Formate	0.011 ± 0.003	0.009 ± 0.001	0.010 ± 0.001	ns	ns

Note: ns *p* > 0.05, * *p* < 0.05, ** *p* < 0.01, *** *p* < 0.001, **** *p* < 0.0001. Red/blue indicates an increase/decrease in the relative metabolite levels, respectively.

## Data Availability

Additional data are provided in the Appendix A.

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
