# Peer review of "Metabolomic Analysis of Trehalose Alleviating Oxidative Stress in Myoblasts"

_ijms, 2023, doi:10.3390/ijms241713346_

Round 1

Reviewer 1 Report

1. Figure 4 and accompanying text.  The design is clear and easy to understand; however, the metabolomics results would be better served showing the comparisons between your four groups using a single PCA to show that there is no effect/overlap when comparing the cells exposed to H2O2 + trehalose and those without one or the other.  Having separate PCAs and OPLSDAs makes it more convoluted and harder to come to any real conclusion.

2. Section 2.6 and Figure 6.  Authors use pathway analysis but don't specify whether all metabolites identified were used for this analysis or if only metabolites of significance were used.  This can make a difference on the quality of the pathway analysis and the relevance on the metabolic pathways of significance.

3. Figure 6. Significance values and consistent markings for Figure 6 should be used.  The graphics show numbers while the legend uses P#.  This should be consistent to avoid confusion.

Author Response

  1. Figure 4 and accompanying text. The design is clear and easy to understand; however, the metabolomics results would be better served showing the comparisons between your four groups using a single PCA to show that there is no effect/overlap when comparing the cells exposed to H2O2 + trehalose and those without one or the other. Having separate PCAs and OPLSDAs makes it more convoluted and harder to come to any real conclusion.

Response: Thank you for your constructive comment. The PCA score plot of the four groups have been added to Figure 4 as Panels A and H, which illustrate the grouping trends of the metabolic profiles of the samples obtained from the trehalose treatment experiment, and those obtained from the trehalose pretreatment experiment, respectively. The legend of Figure 4 has been revised as followed:

 “(A) PCA scores plot of the C, H, T, HT groups for the trehalose treatment experiment. (B, D, F) PCA scores plots for H vs. C, HT vs. H and T vs. C. (C, E, G) OPLS-DA scores plots for H vs. C, HT vs. H and T vs. C. (H) PCA scores plot of the pC, pH, pTH groups for the trehalose pretreatment experiment. (I, K) PCA scores plots for pH vs. pC and pTH vs. pH. (J, L) OPLS-DA scores plots for pH vs. pC and pTH vs. pH.”

Furthermore, the following sentence have been added in the “Result” section of the revised manuscript:

“The PCA score plot of the C, T, H, and HT groups of cells illustrates the grouping trends of the metabolic profiles of these four groups for the trehalose treatment experiment (Figure 4A). Furthermore, the PCA score plots for H vs. C, HT vs. H and T vs. C revealed a clear metabolic distinction among these four groups (Figure 4B, D, F). Similarly, the PCA score plot of the pC, pH, and pTH groups of cells also displays the grouping trends of the metabolic profiles of these three groups for the trehalose pretreatment experiment (Figure 4H), and PCA scores plots for pH vs. pC and pTH vs. pH. revealed a clear metabolic distinction among these three groups (Figure 4I, K).” (Page 6)

  1. Section 2.6 and Figure 6. Authors use pathway analysis but don't specify whether all metabolites identified were used for this analysis or if only metabolites of significance were used. This can make a difference on the quality of the pathway analysis and the relevance on the metabolic pathways of significance.

Response: Thank you for your constructive comment. In this study, we used the concentrations of all identified metabolites to conduct pathway analysis. We have changed the original sentence “Pathway enrichment analysis was conducted to assess the significance of metabolite enrichment within specific pathways.” to  

“Pathway enrichment analysis was conducted to assess the significance of metabolite enrichment within specific pathways based on the concentrations of all identified metabolites.” in the “Materials and methods” section of the revised manuscript (Page 17).

  1. Figure 6. Significance values and consistent markings for Figure 6 should be used. The graphics show numbers while the legend uses P#. This should be consistent to avoid confusion.

Response: Thank you for your constructive comment. We have changed the number # to P# in the revised Figure 6 to keep the consistence between the figure and its legend.

Reviewer 2 Report

This study investigates that trehalose treatment effectively mitigates oxidative stress mediated by H2O2 in C2C12 myoblasts by regulating metabolic profiles, upregulating P62 and Nrf2 proteins, and reducing intracellular ROS. These findings indicate trehalose supplementation as a promising strategy for alleviating oxidative stress in skeletal muscle with potential therapeutic applications. 

The overall concept is intriguing, and the findings hold considerable potential significance. Therefore, with the following minor revisions, this paper can be accepted:

1.     The authors should provide information on the number of replicates for each experiment to assess the statistical significance of the results.

2.     Have the authors investigated the temporal effects of trehalose treatment over longer time points to determine whether the protective effects persist or diminish over time?

3.     The study utilizes C2C12 myoblasts as the model system to investigate the effect of trehalose on oxidative stress in skeletal muscles. While various cell types are known to be susceptible to oxidative stress, do the authors also contemplate the possibility that similar protective mechanisms might extend to other cells? By acknowledging this possibility and discussing it in the context of the broader scientific literature, the study's broader implications could be enhanced, and it could potentially open the avenues for further research and exploration into the protective effects of trehalose against oxidative stress in different cell types as well.

4.     The use of the Nrf2 inhibitor ML385 is an essential part of the study to support the involvement of Nrf2 in trehalose's protective effects. However, the manuscript does not mention whether the use of ML385 had any effect on cell viability independent of Nrf2 inhibition. Certain cell systems have demonstrated sensitivity to ML385 at specific concentrations. Therefore, it would be beneficial if the authors could provide a suitable reference to demonstrate the absence of significant drug toxicity of ML385 at 5 μM concentrations on C2C12 myoblasts.

Author Response

  1. The authors should provide information on the number of replicates for each experiment to assess the statistical significance of the results.

Response: Thank you very much for your constructive comments. In the “Materials and methods” section of the revised manuscript, we have added information about the number of replicates for each group of samples in each experiment.

Cell viability Measurements

“Myoblasts were seeded at a density of 5 × 103 (200 μL per well) in 96-well plates and cultured as described above, with five replicates for each group of samples (n=5).” (Page 13)

Intracellular ROS Measurements

“The cells were then washed with PBS three times and transferred to a black 96-well plate to measure the fluorescence intensity at 525 nm (n = 4).” (Page 14)

Lipid peroxidation assessed by the MDA assay

“The supernatant was transferred to a 96-well plate, and the absorbance at 532 nm and 600 nm was measured (n = 4).” (Page 14)

Assay of cellular T-AOC

“The absorbance at 734 nm was measured (n = 4), and the total antioxidant capacity of the sample was calculated based on a standard curve.” (Page 14)

Extraction of cellular metabolites

“In this study, 10 samples from each of the experimental groups were used for the cell metabolomics analysis.” (Page 14)

  1. Have the authors investigated the temporal effects of trehalose treatment over longer time points to determine whether the protective effects persist or diminish over time?

Response: Thank you for your valuable suggestion. In the “Discussion” section of the revised manuscript, we have added these sentences:

“In addition, in this study, trehalose was added to treat cells either after completion of the H2O2 modelling process, with a treatment time of 24 hours, or during the entire process, with a pretreatment time of 48 hours. Trehalose pretreatment showed a strong protective effect after the third 24-hour supplementation (Figure S1B). Certainly, it is a valuable attempt to investigate the temporal effects of trehalose treatment over longer time points than the third 24-hour supplementation to determine whether the protective effects persist or diminish over time. We will be doing this in the future.” (Page 12)

  1. The study utilizes C2C12 myoblasts as the model system to investigate the effect of trehalose on oxidative stress in skeletal muscles. While various cell types are known to be susceptible to oxidative stress, do the authors also contemplate the possibility that similar protective mechanisms might extend to other cells? By acknowledging this possibility and discussing it in the context of the broader scientific literature, the study's broader implications could be enhanced, and it could potentially open the avenues for further research and exploration into the protective effects of trehalose against oxidative stress in different cell types as well.

Response: Thank you for your constructive comment. It has been demonstrated that trehalose can reduce oxidative stress in other cells or tissues, as reported in the following literature [1-4]. We have added them as References 43-46 together with the following sentence in the revised manuscript:

 “It has been previously demonstrated that trehalose can alleviate oxidative stress in liver cells [43], peripheral blood mononuclear cells [44], the spleen [45], the kidney [46], and so on.” (Page 9)

Literature:

[1]   Bastin, A.R.; Nazari-Robati, M.; Sadeghi, H.; Doustimotlagh, A.H.; Sadeghi, A. Trehalose and N-Acetyl Cysteine Alleviate Inflammatory Cytokine Production and Oxidative Stress in LPS-Stimulated Human Peripheral Blood Mononuclear Cells. Immunological Investigations 2021, 51, 963-979, doi:10.1080/08820139.2021.1891095.

[2]   Honma, Y.; Sato-Morita, M.; Katsuki, Y.; Mihara, H.; Baba, R.; Hino, K.; Kawashima, A.; Ariyasu, T.; Harada, M. Trehalose alleviates oxidative stress-mediated liver injury and Mallory-Denk body formation via activating autophagy in mice. Medical Molecular Morphology 2020, 54, 41-51, doi:10.1007/s00795-020-00258-2.

[3]   Qu, K.-C.; Wang, Z.-Y.; Tang, K.-K.; Zhu, Y.-S.; Fan, R.-F. Trehalose suppresses cadmium-activated Nrf2 signaling pathway to protect against spleen injury. Ecotoxicology and Environmental Safety 2019, 181, 224-230, doi:10.1016/j.ecoenv.2019.06.007.

[4]   Yu, W.; Zha, W.; Peng, H.; Wang, Q.; Zhang, S.; Ren, J. Trehalose Protects against Insulin Resistance-Induced Tissue Injury and Excessive Autophagy in Skeletal Muscles and Kidney. Current Pharmaceutical Design 2019, 25, 2077-2085, doi:10.2174/1381612825666190708221539.

  1. The use of the Nrf2 inhibitor ML385 is an essential part of the study to support the involvement of Nrf2 in trehalose's protective effects. However, the manuscript does not mention whether the use of ML385 had any effect on cell viability independent of Nrf2 inhibition. Certain cell systems have demonstrated sensitivity to ML385 at specific concentrations. Therefore, it would be beneficial if the authors could provide a suitable reference to demonstrate the absence of significant drug toxicity of ML385 at 5 μM concentrations on C2C12 myoblasts.

Response: Thank you for your constructive comment. We determined 5 µM as the concentration of ML385 for treating the C2C12 myoblasts primarily based on those values reported in the literature [5-14] that are shown in the following table. In the revised manuscript, we have added the following sentence in the “Materials and methods” section:

“The concentration of ML385 for treating the C2C12 myoblasts was determined primarily based on those values reported in the literature [65-67].” (Page 17)

Literature

Cell lines

Concentration of ML385

[5]

human lung epithelial BEAS-2B cells

5 mM

[6]

mouse lung epithelial cell line MLE-12

20 μM

[7]

bovine endometrial epithelial cells

20 μM

[8]

Mouse mesangial cells

10 μM

[9]

Human hepatocellular carcinoma cells HepG2

10 μM

[10]

acute myeloid leukemia cell lines HL60, MV4;11, KG1a, U937, HEL, Molm13, NB4, KG1 and Kasumi-1

1.25 / 2.5 / 5 / 10 μM

[11]

mouse neuron cells

7.5 μM

[12]

human lens epithelial cell line B3 cells

5 μM

[13]

rat renal proximal tubular cells NRK-52E

5 μM

[14]

Human gingival fibroblasts

5 μM

Literature:

[5]   Taufani, I.P.; Situmorang, J.H.; Febriansah, R.; Tasminatun, S.; Sunarno, S.; Yang, L.-Y.; Chiang, Y.-T.; Huang, C.-Y. Mitochondrial ROS induced by ML385, an Nrf2 inhibitor aggravates the ferroptosis induced by RSL3 in human lung epithelial BEAS-2B cells. Human & Experimental Toxicology 2023, 42, doi:10.1177/09603271221149663.

[6]   Qiu, Y.-b.; Wan, B.-b.; Liu, G.; Wu, Y.-x.; Chen, D.; Lu, M.-d.; Chen, J.-l.; Yu, R.-q.; Chen, D.-z.; Pang, Q.-f. Nrf2 protects against seawater drowning-induced acute lung injury via inhibiting ferroptosis. Respiratory Research 2020, 21, doi:10.1186/s12931-020-01500-2.

[7]   Fu, K.; Chen, H.; Mei, L.; Wang, J.; Gong, B.; Li, Y.; Cao, R. Berberine enhances autophagic flux by activating the Nrf2 signaling pathway in bovine endometrial epithelial cells to resist LPS‐induced apoptosis. Animal Science Journal 2023, 94, doi:10.1111/asj.13847.

[8]   Tong, J.; Fang, J.; Zhu, T.; Xiang, P.; Shang, J.; Chen, L.; Zhao, J.; Wang, Y.; Tong, L.; Sun, M. Pentagalloylglucose reduces AGE-induced inflammation by activating Nrf2/HO-1 and inhibiting the JAK2/STAT3 pathway in mesangial cells. Journal of Pharmacological Sciences 2021, 147, 305-314, doi:10.1016/j.jphs.2021.08.006.

[9]   Kang, Y.; Song, Y.; Luo, Y.; Song, J.; Li, C.; Yang, S.; Guo, J.; Yu, J.; Zhang, X. Exosomes derived from human umbilical cord mesenchymal stem cells ameliorate experimental non-alcoholic steatohepatitis via Nrf2/NQO-1 pathway. Free Radical Biology and Medicine 2022, 192, 25-36, doi:10.1016/j.freeradbiomed.2022.08.037.

[10] Liu, X.; Zhong, S.; Qiu, K.; Chen, X.; Wu, W.; Zheng, J.; Liu, Y.; Wu, H.; Fan, S.; Nie, D.; et al. Targeting NRF2 uncovered an intrinsic susceptibility of acute myeloid leukemia cells to ferroptosis. Experimental Hematology & Oncology 2023, 12, doi:10.1186/s40164-023-00411-4.

[11] Liu, X.S.; Bai, X.L.; Wang, Z.X.; Xu, S.Y.; Ma, Y.; Wang, Z.N. Nrf2 mediates the neuroprotective effect of isoflurane preconditioning in cortical neuron injury induced by oxygen-glucose deprivation. Human & Experimental Toxicology 2021, 40, 1163-1172, doi:10.1177/0960327121989416.

[12] Ran, H.; Liu, H.; Wu, P. Echinatin mitigates H2O2-induced oxidative damage and apoptosis in lens epithelial cells via the Nrf2/HO-1 pathway. Advances in Clinical and Experimental Medicine 2021, 30, 1195-1203, doi:10.17219/acem/139130.

[13] Dong, W.; Liu, G.; Zhang, K.; Tan, Y.; Zou, H.; Yuan, Y.; Gu, J.; Song, R.; Zhu, J.; Liu, Z. Cadmium exposure induces rat proximal tubular cells injury via p62-dependent Nrf2 nucleus translocation mediated activation of AMPK/AKT/mTOR pathway. Ecotoxicology and Environmental Safety 2021, 214, doi:10.1016/j.ecoenv.2021.112058.

[14] Huang, X.; Liu, Y.; Shen, H.; Fu, T.; Guo, Y.; Qiu, S. Chlorogenic acid attenuates inflammation in LPS-induced Human gingival fibroblasts via CysLT1R/Nrf2/NLRP3 signaling. International Immunopharmacology 2022, 107, doi:10.1016/j.intimp.2022.108706.